# Effect of Quadratus Lumborum Block on Pain and Stress Response after Video Laparoscopic Surgeries: A Randomized Clinical Trial

**DOI:** 10.3390/jpm13040586

**Published:** 2023-03-27

**Authors:** Virna Guedes Alves Brandão, Gustavo Nascimento Silva, Marcelo Vaz Perez, Kai-Uwe Lewandrowski, Rossano Kepler Alvim Fiorelli

**Affiliations:** 1Department of Anesthesiology, Gaffrée e Guinle University Hospital, Federal University of the State of Rio de Janeiro (UNIRIO), Rio de Janeiro 22290-240, RJ, Brazil; 2Department of Surgery and Anesthesia, Federal University of São Paulo (UNIFESP), São Paulo 04021-001, SP, Brazil; 3Center for Advanced Spine Care of Southern Arizona, Tucson, AZ 85712, USA; 4Department of Orthopaedics, Fundación Universitaria Sanitas, Bogotá 111321, DC, Colombia; 5Department of Orthopedics at Hospital Universitário Gaffrée e Guinle, Universidade Federal do Estado do Rio de Janeiro, Rio de Janeiro 20270-004, RJ, Brazil; 6Department of General and Specialized Surgery, Gaffrée e Guinle University Hospital, Federal University of the State of Rio de Janeiro (UNIRIO), Rio de Janeiro 22290-240, RJ, Brazil

**Keywords:** quadratus lumborum block, laparoscopic surgeries, physiological stress response, postoperative pain, postoperative period

## Abstract

Background: There are many surgical and anesthetic factors that affect pain and the endocrine–metabolic response to trauma. The ability of anesthetic agents and neuronal blockade to modify the response to surgical trauma has been widely studied in the last few years. Objective: To evaluate if the anterior quadratus lumborum block contributes to improved surgical recovery, using as parameters analgesia, pulmonary function and neuroendocrine response to trauma. Methods: We carried out a prospective, randomized, controlled, and blinded study, in which 51 patients scheduled for laparoscopic cholecystectomy. Patients were randomly selected and assigned to 2 groups. The control group received balanced general anesthesia and venous analgesia, and the intervention group was treated under general, venous analgesia and anterior quadratus lumborum block. The parameters evaluated were: demographic data, postoperative pain, respiratory muscle pressure and inflammatory response to surgical stress with the plasma dosage of IL-6 (Interleukin 6), CRP (C-Reactive protein) and cortisol. Results: Anterior quadratus lumborum block induced the slowing of IL-6 cytokine production and a decrease in cortisol release. This effect was accompanied by the significant reduction of postoperative pain scores. Conclusion: Anterior quadratus lumborum block is an important strategy for analgesia in abdominal laparoscopic surgery and contributes to reducing the inflammatory response to surgical trauma with an early return of preoperative baseline physiological functions.

## 1. Introduction

The introduction of laparoscopic surgery was an important milestone in the evolution of the surgical treatment of gallbladder pathologies. This technique is now known to shorten recovery and attenuate the inflammatory response resulting from surgical procedures [1]. It is associated with decreased pain scores, wound infection rates, shorter hospital stays, and an earlier return to work [2]. Most cholecystectomies are performed by laparoscopy, this procedure being the most common elective gastrointestinal surgery, with approximately 700,000 procedures performed annually in the United States. 

Video laparoscopy promotes hemodynamic changes by the combination of factors such as pneumoperitoneum, hypercapnia, positioning and anesthetic technique administered to the patient [3]. The use of ideal pneumoperitoneum is vital in safe laparoscopic surgery to ensure visualization of the surgical field and the lower cardiovascular and respiratory repercussions for the patient. However, the increase in intraperitoneal pressure is caused by regional blood flow, generating ischemia and reperfusion after its deflation [4]. 

Although postoperative pain (POP) is less frequent and more easily controlled after minimally invasive procedures than after procedures performed via laparotomy, there is still no consensus on the effectiveness of control measures. With the video laparoscopic technique, the most frequent complaint is related to visceral pain and shoulder pain, resulting from diaphragmatic irritation in pneumoperitoneum caused by CO_2_, which affect 35 to 60% of patients [5]. Pain at the surgical site is one of the factors that also influences the appearance of hypoventilation in the immediate postoperative period. In addition, thoracic and/or abdominal surgeries involve the respiratory muscles, leading to a decrease in strength [6], and are related to hypoxemia and/or acute respiratory failure, with a high prevalence of postoperative complications [7]. 

Three types of pain occur after laparoscopy [8]: visceral pain, abdominal wall pain, and shoulder tip pain. Visceral pain most often occurs as a result of surgical manipulation and irritation of the parietal peritoneum and diaphragm by carbon dioxide gas trapped and dissolved in the abdomen. Placing the patient in the Fowler position during the procedure also contributes to this. A somatic type of pain occurs due to the trocars (surgical instrument) entering the abdominal wall. Pain can also result from electrocautery and irritation caused by peritoneal bile contamination. Previous studies have shown that the inflammation that occurs after cholecystectomy in the gallbladder bed, liver, diaphragm and parietal peritoneum causes pain and nausea [9].

Incisional pain produces a shallow breathing pattern and restrictive dysfunction, which lead to hypoxemia and pulmonary complications. The degree of pain like has an important role in the generation of this cascade. The incisions, the damage to visceras and traction by the pneumoperitoneum lead to inflammatory responses that may affect patients’ postoperative rehabilitation [10].

Trauma triggers an organized tissue response involving the central nervous system, the hypothalamic–pituitary–adrenal axis, and the immune system. Tissue damage is identified by peripheral and central nociceptors that elicit an axonal membrane depolarization response. The electrical impulse generated by the activation of these nociceptors is transmitted by nerve fibers to the dorsal horn of the spinal cord, initiating the activation of the central nervous system and an immediate sympathetic neurological response. After the immediate adrenergic response, different times of systemic endocrine, tissue and hemodynamic response follow [11].

The immune response to trauma is mediated by the perioperative expression of cytokines. Cytokines indicate the activation pathway of the immune response and cell type involved. They are soluble proteins and polypeptides that act as messengers of the immune system. Produced by a variety of cell types, including monocytes, macrophages, lymphocytes, and endothelial cells, they may exert a pro-inflammatory or anti-inflammatory function depending on the effector site. Cytokines are essential for physiological and immunological homeostasis. They are produced in response to a variety of physiological and pathological stimuli. The increase in Interleukin-6 (IL-6) has been associated with an exacerbated response to trauma, being used as a global inflammatory marker. It is believed that IL-6 is the main inducer of acute-phase proteins, such as hepatocyte C-reactive protein (CRP), in addition to causing the differentiation, proliferation and maturation of hematopoietic progenitors [12,13].

IL-6 is a pro-inflammatory cytokine that exerts local and systemic effects, promoting the activation of the immune system, fighting its infection and initiating biochemical mechanisms of healing and tissue repair. It is the main cytokine to be released after surgery and can be considered a reliable marker of the inflammatory response to surgical trauma [13].

It is produced by macrophages and T helper type 2 (Th2) lymphocytes. It activates the inflammatory cascade, acts on the activation of lymphocytes and the differentiation of antibody-producing B cells, promotes the production of acute-phase proteins and has an endogenous pyrogenic effect. In a clinical study carried out to evaluate the relationship between the magnitude of the trauma and the increase in serum IL-6 in an emergency setting, a direct relationship was found between the increase in IL-6, and the degree of trauma, morbidity and mortality. Therefore, dampening the release of IL-6 may mean control the response to trauma [14].

CRP is considered an opsonin and an activator of innate immune cells, particularly neutrophils, in addition to having anti-inflammatory and pro-inflammatory properties. Elevations in CRP levels begin approximately 4–6 h after operative injury and usually peak within 48 h. After uncomplicated operations, its levels begin to decline and usually normalize within 72–168 h. Therefore, it is perhaps not surprising that, given their relationship and plasma kinetics, both IL-6 and CRP seem to similarly reflect the magnitude of surgical trauma.

The response to surgical stress is characterized by the secretion of pituitary hormones and the activation of the central nervous system [11]. Controlling the endocrine response is an important strategy for controlling postoperative outcomes after trauma. Metabolic and hydroelectrolytic changes, resulting from the adrenergic response to the effector endocrine tissue, can precipitate deleterious events in a susceptible organism. Therefore, the use of a multimodal anesthesia, with strategic drugs with different mechanisms of action and regional blocks, is crucial when this objective is being pursued [15,16,17].

There are many surgical and anesthetic factors that affect the response to trauma, and the control of the inflammatory factor is considered of great importance [18]. It has also been postulated that decreasing or abolishing the endocrine metabolic response to the operation may reduce morbidity [10]. The ability of anesthetic agents and neuronal blockade to modify the endocrine and metabolic response to surgical trauma has been widely studied in the last few years [11]. Anesthetic management can affect the immunostimulatory and immunosuppressive mechanisms indirectly by modulating the function of immune cells or directly by the attenuation of the stress response, either by the use of venous agents or regional blocks. Therefore, the type of anesthetic technique may alter the balance between pro- and anti-inflammatory responses, affecting clinical outcomes [19]. The anesthetic blocks allow the blockade of the afferent and efferent sympathetic pathways at relatively low doses, resulting in the profound suppression of hemodynamic and stress responses to surgery. The development of improved recovery protocols in surgery with strategic drugs and regional blockades, aiming to accelerate the return to habitual activities and to decrease the statistics of adverse events in the perioperative and hospital costs, is increasingly necessary [20].

Truncal blocks, as part of perioperative pain management, were introduced into clinical practice over 40 years ago. Initially used with the ilioinguinal–iliohypogastric block and the rectus sheath block, they are more commonly used in the pediatric anesthesia population. Initially, these blocks were performed without ultrasound guidance, using reference techniques. However, the clinical use of truncal block techniques has developed over time and its expansion has been driven by the introduction of ultrasound in the practice of anesthesiology. Although anatomical markers are reliably detected by ultrasound, anterior abdominal wall blocks vary in both the distribution of local anesthetics and the field of coverage. In the search for greater analgesia coverage and a longer duration of postoperative analgesia, fascia transverse plane block and quadratus lumborum block (QLB) were developed [21].

The successful use of QLB with all approaches has been reported in case reports for the following surgical procedures: proctosigmoidectomy, hip surgery, above-knee amputation, abdominal hernia repair, breast reconstruction, colostomy, closure, radical nephrectomy, lower extremity vascular surgery, total hip arthroplasty, laparotomy, and colectomy. Several other cases report a variety of indications for QLB, documenting that sensory blockades include the T7–L2 dermatomes [22].

The objective of this study is to evaluate if blocking in the form of anterior QLB, also known as type 3, contributes in a way which attenuates surgical repercussions, having as its primary parameter analgesia and as its secondary parameters the pulmonary function and neuroendocrine response to trauma.

## 2. Subjects and Methods

The patients were treated in the Deparment of Surgery of the Gaffrée e Guinle University Hospital, Federal University of the State of Rio de Janeiro (UNIRIO). The surgical clinic and anesthesiology departments are accredited by the Federal Ministry of Education and the Brazilian Society of Anesthesiology (SBA). Standard anesthetic techniques and protocols were used. The study was conducted after receiving approval by the Ethics and Research Committee of the mentioned hospital in June 2020 with CAAE nº 26859319.90000.5258 and was registered on the Brazilian clinical trial registration platform (RBR-96xv826).

### 2.1. Sample Selection and Anesthetic Technique Standardization

After applying and signing the informed consent form (ICF), 51 of 56 patients with American Society of Anesthesiology (ASA) I and II were treated by laparoscopic cholecystectomy from October 2020 to June 2021 at the Gaffrée e Guinle University Hospital (Figure 1). Patients were randomized to two groups in a double-blinded manner. Anesthetic blocks were performed by the same researcher (VGAB) who knew the patient’s group at the time of surgery (control or intervention). Randomization was performed using computer-generated numbers. These codes were placed in sealed and numbered envelopes, which were chosen by the surgeon on the day of surgery. The patient was unaware of whether they belonged to the control or intervention group and the participants’ registration data were replaced by codes for the concealment of personal information. The evaluations were performed by another anesthesiologist (GNS) who did not know which technique had been performed. Postoperatively, patients’ pain was assessed with the visual analogue pain scale (VAS). Laboratory and respiratory tests were also performed.

Considering the eligibility criteria of the study, the patients were excluded from the study for: refusal to participate in the study; body mass index (BMI) greater than or equal to 40; smoking history or obstructive pulmonary disease, presence of peripheral neuropathies; coagulopathies or hypersensitivity to drugs used for analgesia; infection at the puncture site; cholecystitis, deformities or previous spinal operations; dementia or other states that would prevent the adequate understanding of the use of the numeric VAS pain scale; immunological diseases, malignant neoplasia, use of opioids or anti-inflammatory drugs in the preoperative period; use of antidepressants and anticonvulsants; and requirement of conversion of surgery to conventional technique (open surgery).

All patients upon admission to the operating room were monitored according to the recommendations of the ASA and received peripheral venous access with an 18 or 16 intravenous catheter. All surgeries were performed under balanced general anesthesia; both groups received routine venous analgesia. The intervention group additionally received ultrasound-guided anterior QLB, which was performed with the patient anesthetized soon after the induction of general anesthesia. The anesthetic drugs and their dosages were standardized so that there were no differences between the groups. The induction of general anesthesia involved the following: propofol 2.5 mg/kg, rocuronium 0.5 mg/kg and fentanyl 5 mcg/kg. Intraoperative maintenance involved the following: sevoflurane 1–2 CAM with oxygen 50% and air 50%. All patients received dipyrone at a dose of 30 mg.kg−1 intravenously and ketoprofen 100 mg EV at the end of surgery, in addition to infiltration of the surgical portals with ropivacaine 0.3% volume of 10 mL. In the immediate postoperative period, all patients received intravenous ketoprofen 100 mg in regular doses of 12/12 h, and only received common analgesics (intravenous dipyrone, 30 mg/kg/dose 6/6 h) or opioids (50 mg tramadol, intravenous, 6/6 h) if pain was greater than 5. Ondansetron 4 mg was administered intravenously in the intraoperative period for the prophylaxis of nausea and vomiting. Surgical time was similar between the groups, lasting less than 3 h.

### 2.2. Anterior Quadratus Lumborum: Considerations and Block Description

Ultrasound-guided quadratus lumborum block (QLB) is a recently described fascial plane block in which the anesthetic is injected adjacently to the quadratus lumborum (QL) muscle with the goal of anesthetizing the nerves in the thoracolumbar region [16]. It is believed that the anesthetic, injected anteriorly into the QL muscle and posteriorly into the transversalis fascia, will disperse into the paravertebral space, travelling posteriorly and laterally to the arcuate ligaments of the diaphragm along the endothoracic fascia in order to block the somatic nerves and the sympathetic trunk according to the level of dispersion of the anesthetic. Anterior QLB can generate analgesia from T10 to L4. For subcostal QLB (Anterior QLB subtype), local anesthetic, injected anteriorly into the QL between the QL muscle and into the anterior layer of the thoracolumbar fascia, caused cephalad dispersion near the T12 rib with the displacement of the anterior lamina of the thoracolumbar fascia. This produces reliable dermatomal coverage from T6–T7 to L1–2 [23]. 

As QLB is a classic method of injecting intramuscular (interfascial) medication, the possibility of infection is much lower than when performing neuraxial blocks. To date, no infections have been described as a result of QLB. The advantage of the QLB over other abdominal wall blocks is the fact that the passage of the needle and the site of application of the local anesthetic are far from the peritoneal cavity, the visceral abdominal organs and the large blood vessels. Therefore, needle trauma is minimized here in terms of unintentional puncture of the peritoneum, intestine, liver, kidney, and large blood vessels associated with blind (non-ultrasound) methods of TAP and II-IH block performance. Performing QLB under ultrasound control, with mandatory monitoring of the needle tip carried out before drug injection, significantly increases the level of safety and efficiency of the technique. There are no data on neurological damage, as the local anesthetic is not injected in the immediate proximity of the large nerve but is injected into the space which is rich in small nerve endings. Therefore, it is generally accepted that QLB can be performed under general and regional anesthesia [24]. The current literature on QLB reports the use of 4 different approaches, with authors using varying nomenclatures to describe each block. It has been classified based on the anatomical location of the needle tip in relation to the muscle. Thus, the following terminology is adopted: posterior, lateral, anterior, and intramuscular approaches [22].

In this study, the anterior or type 3 QLB was performed right after anesthetic induction. A low-frequency curvilinear transducer was used to facilitate tissue penetration of the ultrasound and the establishment of a wide field of view. With the patient in lateral decubitus, the transducer is placed in transverse orientation on the posterior or midaxillary line at the L2–L4 level in order to visualize the QL and psoas muscle. To identify the QL muscle, it is important to observe some structures as reference, such as the aponeuroses of the abdominal wall muscles (external oblique, internal oblique and transverse abdominal) which are located posterolaterally to the muscle. The QL muscle is often hypoechoic in relation to the psoas major muscle, which is located anteromedially. Additionally, the lumbar transverse processes are apparent due to their curved hyperechoic appearance.

The recommended dose of local anesthesia varies from 0.2–0.4 mL/kg of 0.2–0.5% Ropivacaine or of 0.1–0.25% Bupivacaine. The dose was adjusted respecting the toxic dose of the local anesthetic and the anterior QLB performed bilaterally. In this study, 20 mL of 0.3% Ropivacaine was administered bilaterally (total of 120 mg) using a Stimuplex A100 needle.

### 2.3. Pain Evaluation

The efficacy of analgesia produced by the two techniques studied was evaluated at regular intervals by the numeric verbal rating scale (NVRS). Patients were asked to answer the question, “Are you feeling pain? How severe is the pain on a scale of 0–10 (0—no pain; 10—the most severe pain)?”. The intensity of pain was measured 1 h, 4 h and 24 h postoperatively by this scale, and the consumption of pain killers (tramadol) was also recorded. Categories were constructed based on the intensity of pain reported by the patients, with the value 0 being considered Painless; 1–3 being considered “Light”; 4–6 “Moderate”; and 7–10 “Severe” pain.

### 2.4. Laboratory Assays

For the determination of the markers of inflammatory response measuring IL-6, CRP and cortisol, samples were collected at anesthetic induction (preoperative) and at 4 h and 24 h after the surgical procedure had been performed. The blood samples of the research participants were collected in a tube without additives and the serum was preserved at −112 °F until the moment of analysis without freezing and thawing cycles. The trials were conducted in the laboratory of the National Program for Quality Control (NPQC), in the state of Rio de Janeiro. The research participant did not know the analgesia technique performed in each procedure. IL-6 serum concentration was determined by the electrochemiluminescence immunoassay (ECLIA) using the Roche Cobas e411 immunoassay analyzer. CRP serum concentration was determined with the immunoturbidimetric assay using the Bioclin 3000 automated analyzer. Cortisol serum concentration was determined with the chemiluminescence immunoassay (CLIA) using the Abbott Architect i1000 immunoassay analyzer. Statistical analysis was performed during the process of outcome assessment, and the research participant did not know the analgesia technique performed in each procedure.

### 2.5. Evaluation of Respiratory Muscle Strength

The evaluation of respiratory muscle strength, maximum inspiratory pressure (MIP) and maximum expiratory pressure (MEP) was performed using an analog manovacuometer. Individual rubber ducts were used with a rigid-plastic diver-type mouthpiece with an internal diameter of 2 mm and a length of 15 mm. This was performed according to the method recommended by Black and Hyatt [25]. Evaluations were performed on the day of admission (preoperative), as well as 4 h and 24 h after the procedure.

The measurements were performed with the patient in a “beach chair” position (headboard elevated 45 degrees). Initially, they were oriented to breathe closely to the tidal volume and, after three breaths, patients were asked to perform a maximum forced expiration (residual volume) and then a maximum static inspiratory effort. These methods required being sustained for 3 s, with nasal occlusion, in order to obtain the MIP. Subsequently, to measure the MEP, the patient breathed at a level close to the tidal volume for three cycles and performed a maximum inspiration effort (total lung capacity), followed by a maximum static expiratory effort sustained for 3 s [26]. Both maneuvers were repeated three times and paired with a one-minute interval in which the highest value obtained was recorded. For both maneuvers, the patients took three measurements for improved reproducibility. All measurements were made by the same researcher, who was careful to maintain the same pattern of verbal commands.

### 2.6. Statistic Analysis

In the present study, 56 patients were recruited. After applying selection criteria, 51 patients were included into the analysis. One participant was excluded because of spinal deformity, two were excluded because of high BMI > 40 and two others refused to take the respiratory tests. 

Study power analysis was calculated post hoc. Power calculation was performed for the variables of Pain 24 h after procedure and MEP 4 h after procedure, which showed the statistical significance between groups. The test power for the Pain 24 h was equal to 0.962. As for the MEP 4 h, the resulting power was 0.773. Thus, it is considered that the power of the study was 0.773, considering the minimum power calculated, and a confidence level of 95% was attained (α = 0.05).

The *t* test was chosen for quantitative variables whose data presented normal distributions through the Shapiro test, both for the control and intervention groups, and where the variances between the groups were considered homogeneous when applying the Bartlett test. The central trend measure used to perform the *t* test is the mean of the groups.

For the quantitative variables that, through the Shapiro test, did not present normal distributions for the control and intervention groups, the test chosen was the Mann–Whitney nonparametric test. The central trend measure used to perform the Mann–Whitney test was the median of the groups.

In considering the qualitative or categorical variables, the test chosen was the non-parametric Chi-square test, which takes into account that the control and intervention groups are independent.

The R software 4.0.3 version was selected to perform the hypothesis tests between the patients in both groups, and the Mann–Whitney test was applied to compare each variable between the groups.

## 3. Results

There were 51 patients enrolled in the study. The two groups were similar regarding age, BMI and comorbidities. No complications were observed, and there were no mortalities (Table 1).

### 3.1. Postoperative Pain and Opioid Usage

A chi-square hypothesis test was performed for the POP variable, and the VAS of pain score 24 h after the procedure. There was a statistical difference between the control and intervention groups (*p* value = 0.0008). There was no statistically significant difference in the VAS scores obtained 1 h and 4 h after procedure between the control and intervention groups (Table 2).

We found no significant difference regarding the opioid use between the groups.

### 3.2. Interleukin 6 (IL-6)

When analyzing the three time points in both groups (before the procedure, 4 h and 24 h after), IL-6 presented lower values preoperatively, but 4 h after, IL-6 levels were higher than 24 h after the procedure. Within the control group, there was a high variability of IL-6 for patients 4 h after the procedure. The dispersion values calculated for all the groups was 1.0146, with a value of 1.1130 for control, and for intervention the value was 0.8578 (Table 3).

### 3.3. Cortisol

Observing the median values for the cortisol variable, the highest median found was 22.2 for the control group at 4 h after the procedure. The lowest median found was 9.8 for the intervention group measured preoperatively. According to the *p* values presented in Table 4, there was no statistical difference for the times before the procedure, 4 h after the procedure or 24 h after the procedure for the cortisol variable.

### 3.4. C-Reactive Protein (CRP)

The measurement of CRP medians during the preoperative and postoperative periods did not show many alterations between the intervention and the control group, having the following values: before the procedure (Control = 2/Intervention = 2); 4 h after procedure (Control = 3/Intervention = 4); and 24 h after procedure (Control = 36.15/Intervention = 39.65) (Table 5).

### 3.5. Respiratory Function

We observed that the MEP measurement, 4 h (Control = 47.5/Intervention = 65) and 24 h (Control = 62.5/Intervention = 70) after the procedure showed a greater difference between the medians of the intervention in relation to the control Group compared to the preoperative period (Control = 85/Intervention = 80). Similarly, the MIP measurement, 4 h (Control = 35/Intervention = 40) and 24 h after the procedure (Control = 45/Intervention = 55) showed a difference between the medians of the intervention compared to the control group in relation to the preoperative period (Control = 67.5/Intervention = 60) (Table 6).

## 4. Discussion

The triad of hypnosis, immobility and antinociception, necessary at the moment of anesthesia, is achieved through the synergism of different medications. General anesthesia can inhibit the activity of the central nervous system, reducing surgical trauma to the body, but it has no significant inhibitory effect on the noxious stimulation signal of the somatic nerve or on ascending sympathetic nerve transmission. 

Strategies to control the neuroinflammatory response to trauma are needed, among which pharmacological interventions are required, especially minimally invasive surgery and neural block techniques [27]. This study evaluated a relatively new neural block technique, which is the anterior QLB approach with promising analgesic coverage, which has been demonstrated in previous clinical trials of local anesthetic dispersion [28,29]. Currently, QLB is performed as one of the perioperative pain control procedures for all generations (pediatrics, pregnant women and adults) in abdominal surgery. However, disagreement persists over the best approach with which to administer the blockade, the mechanisms action, and the nomenclature [28,30]. 

Anterior QLB in this study was performed after the induction of anesthesia due to the time of the plasma peak of Ropivacaine, which is on average 40 min, aiming for greater safety and greater efficacy of the blockade at the end of the procedure. The blockade was performed bilaterally, requiring careful consideration of patient positioning and the calculation of the toxic dose of the local anesthetic. 

The fascial planes in the abdominal compartment follow the QL and PM, medially and laterally, through the arcuate ligaments and the aortic hiatus of the diaphragm, forming the endothoracic fascia. This provides a potential route of dissemination of the local anesthetic from the abdominal cavity to the thoracic cavity and paravertebral space, thus achieving clinical effects [31,32]. In addition to serving as a conduit for local anesthetic spread into the thoracic paravertebral space, the thoracolumbar fascia, which has a high-density network of sympathetic fibers as well as mechanoreceptors, is believed to be another major component responsible for the QLB effects [33].

The majority of authors agree that QLB has an outstanding analgesic effect on pain, reducing it to scores of 1–2/10 as assessed by VAS, and this usually lasts for more than 24 h. Patients who receive QLB as part of postoperative pain therapy have lower pain levels both when resting and moving, which is important for early mobilization. The analgesic effect is as good as the one achieved by opioids, and there are no unwanted opioid effects such as nausea and vomiting [34]. 

According to prospective studies published by Blanco, Ansari & Girgis [28], in 2015 and 2016, the need for morphine has been significantly reduced postoperatively in patients who received paracetamol, NSAID, and QLB as part of the multimodal postoperative analgesia compared to patients who received only paracetamol and NSAID but did not receive QLB. Comparative studies have shown that the QLB covers a topographically broader field (Th7–Th12, compared to TAP Th10–Th12) and yields a prolonged pain-free condition compared to the TAP block (24–48 h for QLB versus 8–12 h for the TAP block) [35,36,37]. 

QLB provides early and rapid pain relief in a high percentage of patients and allows early ambulation, which is one of the most important measures in the prevention of deep vein thrombosis and thromboembolic complications [20].

Pain is entirely subjective and its links with pathology are indirect; the only way to successfully assess pain is to believe the patient. In this study, pain was evaluated utilizing the VAS. This method is more preferred by patients for its simplicity as well as in its greater sensitivity in comparison to other pain scales in calculating the pain intensity changes that occur [38,39].

Metanalysis exhibited several superiorities of QLB for patients undergoing laparoscopic surgeries. The results indicated that application of QLB was associated with a smaller number of patients requiring additional analgesia, with reduced intraoperative opioid consumption and postoperative opioid consumption, and with lower incidences of postoperative nausea and vomiting (PONV) compared to placebo [38]. In contrast, Vamnes, Sorenstua, Solbakk, Sterud, and Leonardsen [39] concluded that preoperative anterior QLB for laparoscopic cholecystectomy does not affect postoperative opioid requirements and pain, but that it may decrease PONV. 

This study demonstrated a significant reduction in pain scores and opioid consumption in the intervention group (*p* value < 0.05) and, as demonstrated in two randomized controlled trials, that QLB reduces cumulative opioid consumption for 48 h after caesarean section [40,41]. Santos, Rabelo, Borges, Silva, and Silva [42] observed that, after laparoscopic cholecystectomy, there was significant reduction in respiratory muscle strength on the first postoperative day in relation to the preoperative period, with reductions in the MIP and MEP, despite the laparoscopic surgery causing less pulmonary compromise than conventional surgery. Our study showed less reduction in pulmonary pressures in the intervention group in relation to the control group at 4 and 24 h after surgical procedure. Patients treated with the anterior QLB recovered preoperative muscle strength early, which is an important result, since anesthesia aims at reducing the repercussions of surgical trauma and facilitating an early return to function. The early recovery of lung function in patients undergoing QLB may be related to the optimization of analgesia with this technique.

During surgery, an immune/inflammatory response is initiated, determining varying degrees of clinical implications. A study of elderly patients concluded that QLB could improve postoperative cognitive function in this group undergoing laparoscopic radical gastrectomy. This may be related to the suppression of the inflammatory response after surgery. Compared with the control group, HMGB1, TNF-α and IL-6 levels were significantly decreased 1 and 3 days after surgery in the intervention group (*p* < 0.05) [43]. Our research team carried out systematic review of the main biomarkers related to the inflammatory response to surgical trauma in order to define the most sensitive and specific marker for this research. We did not find in the literature a protocol for evaluating the inflammatory response to surgical trauma. Thus, we emphasize the importance of standardizing dosages and collection intervals for future research.

According to the literature, CRP was the most described biomarker, followed by the IL-6 and TNF-a responses to surgical intervention. IL-6 rose 1 h after surgery with a plasma peak at 4 to 6 h. CRP starts its rise from 4–6 h (induced by IL-6) with a plasma peak at 48 h. Although both IL-6 and CRP reflect similarly the magnitude of trauma, the kinetics of these biomarkers are not identical. When analyzing the IL-6 variable at the three time points, a high variability among patients 4 h after the procedure was observed within the control group. It was confirmed via the coefficient of variation calculations that the dispersion of the IL-6 variable was greater in the control than in the intervention group. This reduction in the coefficient of variation may demonstrate an attenuation of the immune/inflammatory response to surgical trauma in the group submitted to anterior QLB. In a study by Zhu, Qi, He, Zhang, and Mei [43], a significant reduction in IL-6 values was demonstrated in elderly patients.

Regarding CRP levels between the control and intervention group, we noticed similar level patterns without many alterations at the times evaluated. In the intervention group, there was only a change in plasma kinetics for IL-6, which did not occur with CRP. The duration of the effect of anterior QLB is about 12–24 h, having a greater effect the IL-6 kinetics [44]. Cortisol and ACTH intermediates are activated by IL-6 secretion and are considered acute-phase hormonal reagents to surgical stress. Cortisol secretion is also associated with the severity of trauma and stress response. Cortisol is expected to continue to increase many days after surgery, with peak levels approximately 4–6 h after the incision [45]. Our study showed lower cortisol values within the first 4 h for the intervention group. Thus, anterior QLB seems to interfere in the hormonal response to surgical stress in the first 4 h. Significant results were found for reduction in pain scores (*p* < 0.05) in the intervention group. The data and dosage of several substances and cellular components, including some hormones, leukocyte count, inflammatory cytokines and analysis of T-lymphocyte behavior, may be useful in monitoring for systemic inflammatory syndrome (SIS) after elective surgeries. However, there are divergences in the results of some studies, which can be attributed to potential confounding factors related to the moment of collection and mediator-dosed, intrinsic factors related to the patients, and the peculiar behaviors of the mediator to be studied.

In our study, the exclusion of ASA Status III patients who have severe systemic disease with the functional limitations of advanced age, obesity, emergency situations, and autoimmune diseases and conversion to open surgery were necessary, because such conditions can alter the inflammatory response. These comorbidities can interfere with the measurement of biomarkers. The surgical time is directly related to the magnitude of the trauma. Therefore, we limited the search to elective laparoscopic cholecystectomies that lasted an average of 2 to 3 h. The team was the same and the same researcher performed all anesthetic blocks due to their operator-dependent procedures.

Our study had limitations: for the CRP parameter. A longer patient follow-up would be necessary since the plasma peak of this marker occurs 48 h after the trauma. The cortisol result may also have interfered with the surgical schedules. Our research was carried out during the COVID 19 pandemic, when elective surgeries were reduced in hospitals. The relatively small sample of patients may also have hampered our analysis.

## 5. Conclusions

There are several studies in the literature which evaluate the QLB effect on the pain and surgical stress responses. The present clinical trial evaluated the effect of anterior QLB on the inflammatory response. Our results suggest a reduced inflammatory response to surgical trauma in cholecystectomy patients. This technique can be an important adjunct to multimodal anesthesia in minimally inivasive surgical intervention. Additional applications have been suggested; namely in the upper and lower abdomen and pelvic cavity. QLB optimizes the control of the neurohumoral response while maintaining excellent lung function and pain control.

There is great importance of controlling the metabolic and inflammatory response to surgical trauma. The paradigm shift currently playing out is from patients having essentially been treated for pain with analgesics, to treating them with neuroinflammatory treatment. In fact, pain and surgical stress response are closely related. The method described by the authors of this article., expands the range of therapeutic interventions and perioperative care available to physicians, delivering improved patient satisfaction and impacting public health outcomes.

## Figures and Tables

**Figure 1 jpm-13-00586-f001:**
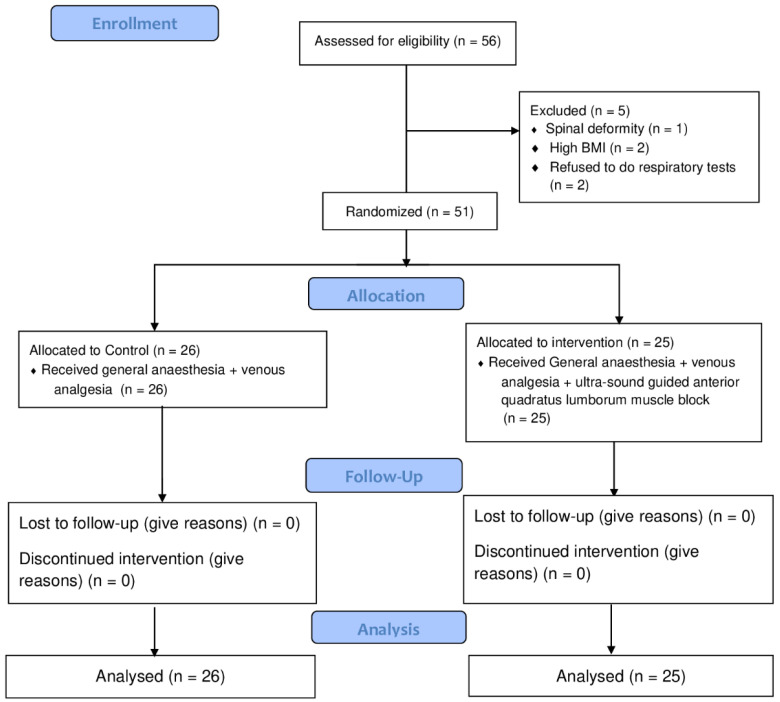
CONSORT study flow diagram.

**Table 1 jpm-13-00586-t001:** Demographics data of the study. BMI = body mass index; ASA = American Society of Anesthesiology Status; SD = standard deviation; * chi-square test, ** Mann–Whitney test.

	Patients (*n* = 51)	Control (*n* = 26)	Intervention (*n* = 25)	*p* Value
**Age (years)**				
mean (± SD)	50.5 (±13.7)	53.2 (±13.0)	47.7 (±14.1)	0.163 **
**Sex** *n* (%)				
Female	38 (74.5)	18 (69.2)	20 (80.0)	0.575 *
Male	13 (25.5)	8 (30.8)	5 (20.0)	
**BMI (kg/m^2^)**				
mean (± SD)	26.9 (±6.5)	27.46 (±7.1)	26.4 (±6.2)	0.888 **
**ASA** *n* (%)				
ASA I	18 (35.3)	8 (30.8)	10 (40.0)	0.692 *
ASA II	33 (64.7)	18 (69.2)	15 (60.0)	

**Table 2 jpm-13-00586-t002:** POP Values. POP = postoperative Pain; * chi-square test.

POP	Control (*n* = 26)	Intervention (*n* = 25)	*p* Value *
1 h after procedure	2/4/10/10	4/11/5/5	0.0644
Painless/Light/Moderate/Severe			
4 h after procedure	7/12/7/0	11/13/1/0	0.0668
Painless/Light/Moderate/Severe			
24 h after procedure	11/14/1/0	23/2/0/0	0.0008
Painless/Light/Moderate/Severe			

**Table 3 jpm-13-00586-t003:** IL-6 values. IL-6 = Interleukin 6; IQR = interquartile range; * Mann–Whitney test.

IL-6 Median (IQR) pg/mL	Patients (*n* = 51)	Control (*n* = 26)	Intervention (*n* = 25)	*p* Value *
Before the procedure	1.6 (0.9)	1.7 (0.8)	1.5 (1.2)	0.952
4 h after procedure	59.9 (76)	65.8 (81.8)	59.9 (63.7)	0.744
24 h after procedure	21.6 (28.5)	23.6 (26.1)	20.1 (29.1)	0.877

**Table 4 jpm-13-00586-t004:** Cortisol Values. IQR = interquatile range; * Mann–Whitney test.

Cortisol Median (IQR) mcg/dL	Patients (*n* = 51)	Control (*n* = 26)	Intervention (*n* = 25)	*p* Value *
Before the procedure	10.4 (8.1)	10.7 (10.4)	9.8 (5.9)	0.720
4 h after procedure	20.7 (19.8)	22.2 (16.95)	18.5 (17.7)	0.205
24 h after procedure	13.3 (9.7)	11.8 (9.7)	14.4 (8.9)	0.275

**Table 5 jpm-13-00586-t005:** C-Reactive Protein *t* test Results. SD = Standard Deviation.

Variable Mean (SD)	Control	Intervention	*p* Value
CRP mg/L			
Before procedure	2.00 (2.90)	2.00 (6.52)	0.617
4 h after procedure	3.00 (4.24)	4.00 (5.60)	0.962
24 h after procedure	39.48 (22.77)	45.94 (22.45)	0.327

**Table 6 jpm-13-00586-t006:** MEP and MIP *t* test results. SD = standard deviation.

Variable Mean (SD)	Control	Intervention	*p* Value
MEP			
Before procedure	86.92 (17.89)	81.00 (19.36)	0.262
4 h after procedure	49.23 (19.01)	63.80 (23.01)	0.017
24 h after procedure	62.50 (21.64)	70.00 (20.36)	0.124
MIP			
Before procedure	69.23 (19.42)	61.20 (14.16)	0.097
4 h after procedure	35.00 (18.86)	40.00 (14.49)	0.166
24 h after procedure	45.00 (19.66)	55.00 (12.88)	0.050

## Data Availability

The data are contained within the article.

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
