# Peer review of "Effect of Quadratus Lumborum Block on Pain and Stress Response after Video Laparoscopic Surgeries: A Randomized Clinical Trial"

_jpm, 2023, doi:10.3390/jpm13040586_

Round 1

Reviewer 1 Report

Dear Authors, 

Comments 

Major: 

1. Methodology. Were the elective cases only? This should be clarified.

2. Blindness of all research participants should be clarified. It was written that statistician and patients were blinded to the study group assignment. What about other investigators? Who assessed pain, who performed respiratory tests? Respiratory tests were performed at the same time points as pain assessments? In what order? As pain scores may be affected by respiratory efforts.

3. Statistical analysis. Power calculation is very unclear…Was it made post-hoc? Or before the study? On what parameters power analysis was based?

4. Statistical description should include the following: description of test for data normality check, what statistical tests were used to compare main and secondary outcomes. What central and precision statistical parameters  are used for data presentation. All these are absent in the statistical description.

5. Results. Except for demographic variables, results are not presented properly:

P values for pain scores are not supported by central and precision numbers. Differences in medians or means not presented. This is further obscured by figures: pain scores in figures actually do not represent p values in the text?? When no other information is provided in the text, there is no evidence of the reported statistical differences.    

6. Nor p values, nor precision values are provided for opioid consumption.

7. No precision values nor p values are provided for any other of reported outcomes. No units are presented for laboratory outcomes (CRP, IL-6, cortisol).

8. The label of figure 4 reads: "Figure 4. Boxplots and Graphs from the Interleukin 6 (IL-6) variable per patients in the control and 291 intervention groups." However, instead of IL-6 pain scores are presented.

Other comments

1. Introduction.

Introduction must include review on the use QLB for cholecystectomy, as many articles are published by now, clearly pointing what new information does the present paper adds. Descriptions of the QLB technyques can be omitted as these are in detail described elsewhere. Intro must include the updated info on the current research on stress hormones and regional blocs in surgery and what new aspects will be studied in presented research.

3. All figures need to be elaborated – y axis labeled, descriptive statistics of the presented data provided.

4. Language needs extensive English corrections.

As the description of the results is very obscured, the Discussion and Conclusion sections should be revised after clear presentation of the results of the study.

In conclusion, an article needs major and extensive corrections. I would like to note that reporting the results in double blinded randomized trials is a key stone and Consort requirements should be followed and met. 

Author Response

Dear Reviewer,

We are glad to know that, based on the advice received, our manuscript entitled “Effect of Quadratus Lumborum Block on Pain and Stress Response after Video Laparoscopic Surgeries: a randomized clinical trial" has merit for publication. We thank you for your time spent evaluating our manuscript and for the valuable input given to improve it.

We send a marked-up copy of our revised manuscript (RM) that highlights changes made to the original version. The corresponding questions raised by you can be found highlighted and italized, our response following it, detailing point-by-point, in the document anexed.

Reviewer 2 Report

I was glad to review the work of the authors regarding this very interesting article on the Effect of Quadratus Lumborum Block on Pain and Stress Response after Video Laparoscopic Surgeries.

Despite the major advances in anesthesiology and pain management, there are still numerous unanswered questions regarding the ideal technique to achieve better pain control for patients undergoing laparoscopic surgery

Generally, the manuscript is well-written and the topic is very interesting.

Nevertheless, the manuscript requires to be improved to ameliorate the introduction and discussion.

In general, the Manuscript may benefit from several minor revisions, as suggested below:

1) "For determination of markers of inflammatory response measuring Interleukin 6 (IL- 6), C-reactive Protein (CRP) and cortisol, samples were collected at anesthetic induction (preoperative), 4 hours and 24 hours after the end of surgery."

Why did you choose these inflammatory markers? Why not white blood cells, neutrophils, or procalcitonin? Why not have these tests 12 hours postoperatively? 

2) Could you please explain why patients with ASA III were excluded?

3) "The two groups were similar regarding age, BMI, and comorbidities" I would suggest adding a p-value in the table1.

4) Did all patients undergo laparoscopic cholecystectomy because of cholelithiasis or were there cases of chronic cholecystitis or other diagnoses?

5) What was the mean operating time of each group? Did you have a significant difference?

6) In your study the visual analogue scale-VAS was used. Why did you choose this scale and not the Numeric Pain Rating Scale (NPRS)? I would like a brief discussion on pain rating scales that are used up to date in recent studies.

"According to the literature, NRS has been chosen in a randomized prospective trial because compared to other pain intensity scales it is preferable by patients, as well as in comparison to other pain scales (such as the Visual Analogue Scale, VAS), it is more sensitive in calculating the pain intensity changes that occur"

I would suggest adding this information to your discussion section

https://pubmed.ncbi.nlm.nih.gov/33155461/

Author Response

Dear Reviewer,

We are glad to know that, based on the advice received, our manuscript entitled “Effect of Quadratus Lumborum Block on Pain and Stress Response after Video Laparoscopic Surgeries: a randomized clinical trial" has merit for publication. We thank you for your time spent evaluating our manuscript and for the valuable input given to improve it.

We send a marked-up copy of our revised manuscript (RM) that highlights changes made to the original version. The corresponding questions raised by you can be found highlighted and italized, our response following it, detailing point-by-point, in the document attached.

Reviewer 3 Report

Thank you for giving me the opportunity to review this manuscript. The study is very interesting and well structured. I have only two small perplexities: how did you evaluate the intraoperative analgesia and how did you choose the statistical test used for the comparison.

Author Response

(The authors gave the same response as above.)

Round 2

Reviewer 1 Report

2023.02.22.

The article did not improve significantly and many remarks were not addressed:

Introduction

The relevant new information regarding latest research on the influence of regional blockade on neurohumoral stress response is not provided. The novelty of the study is not provided.

Serious methodological drawbacks:

Exclusion  criteria: authors do not mention smoking history or obstructive pulmonary disease, which are relevant in pulmonary tests.

Post-hoc power analyses in a randomized trials is serious methodological drawback.

Other:

QLB block technique should include description how it was done. Instead authors describe how it should be done (lines 255 – 323).

Figures 2 and 3 are unnecessary.

Results

Not a single table is cited in the text.

Primary outcome variables not enough described and difficult to understand : lines 435-442 – what was compared? : Continuous pain scores? Or pain severity groups (1-4)?

Lines 440-443 doesn’t make sense….

Measurement unit for IL-6 not presented.

Figure 5 is not improved: pain scores represented instead IL-6.

Precision values for C-reactive protein not provided.

Authors write: “lower values for intervention groups. 514 Comparing the median measurements for the cortisol variable, the greatest 515 difference between the medians of the groups is in the time 4 hours after the procedure 516 (Ccontrol group = 22.18 / Iintervention group = 18.51). For the other times, the 517 measurements before the procedure (Ccontrol group = 10.71 / Iintervention group = 9.76) 518 and 24 hours after (Ccontrol group = 11.84 / Iintervention group = 14.37) showed 519 differences between medians smaller than those for the time 4 hours after the procedure.” P values show that there was no difference between the two groups. Such writing style is unacceptable.

Lines 454-457 repeat methodology.

Figures are not enough improved (placing intervention and control group on the same graph would be more illustrative). Figure 3 or 4??? Impossible to understand where it ends? Total mixup... Information provided in the pictures and Tables is redundant.

Lines 483-488 doesn’t make sense…

Respiratory function : description of results is unclear.

After a very unclear presentation of results, it doesn’t make sense reading the discussion.

The authors lack basic knowledge and experience in scientific writing. First of all, this article should be revised by experienced scientific writers or colleagues and only after that presented to the scientific journal. It is still a draft that needs extensive corrections.

Author Response

Dear Reviewer,

We acknowledge our difficulties and limitations in providing the best report for our trial. For this last review, we tried our best to deliver the data and information required by you. It may had some minor miscomprehensions along the way, but we believe we improved the article considerably.

We submitted one last time the revised marked up copy of the manuscript highlighting the alterations and a PDF file, as well.

Our article was revised by a researcher with experience in writing in scientific style for journals. Please, see the attachment.

We thank you for your considerations, we revised the manuscript the best way we could, but we have nothing more to add or to revise further to this point.

Best regards,

Virna Guedes Alves Brandão.
